# Evaluating the Feasibility and Reliability of Remotely Delivering and Scoring the North Star Ambulatory Assessment in Ambulant Patients with Duchenne Muscular Dystrophy

**DOI:** 10.3390/children9050728

**Published:** 2022-05-16

**Authors:** Nicholas Emery, Kate Strachan, Richa Kulshrestha, Jan Herman Kuiper, Tracey Willis

**Affiliations:** 1Neuromuscular Service, TORCH Building, Robert Jones and Agnes Hunt Orthopaedic Hopsital, Shropshire SY10 7AG, UK; nicholas.emery1@nhs.net (N.E.); kate.strachan@nhs.net (K.S.); jan.kuiper@nhs.net (J.H.K.); tracey.willis1@nhs.net (T.W.); 2School of Pharmacy and Bioengineering, Keele University, Staffordshire ST5 5BG, UK

**Keywords:** Duchenne muscular dystrophy, north star ambulatory assessment, video assessment

## Abstract

Objective: The North Star Ambulatory Assessment (NSAA) is a validated 17-item functional rating scale and widely used to assess motor function in boys with Duchenne muscular dystrophy (DMD). The SARS-CoV-2 pandemic and subsequent Government ‘lockdown’ resulted in no face-to-face clinic visits hence the motor abilities were not monitored. The aim was to investigate whether the NSAA was feasible and reliable by video assessment. Method: Ten ambulant DMD boys were selected from the electronic hospital records. Two physiotherapists scored the boys’ NSAA independently and the intraclass correlation coefficient was used to assess agreement. The video scores were compared to two previous NSAA in-clinic scores. Results: Mean scores (SD) for clinic visit one were 22.6 (4.19) and clinic visit two 21.8 (5.3). The two physiotherapists video mean scores were 20.6 (5.66) for physiotherapist 1 and 20.6 (6.53) for physiotherapist 2. The intraclass correlation coefficient was 0.98 (95% CI 0.93–1.00) for the total NSAA and 1.00 (95% CI 1.00 to 1.00) for the rise time. The mean decline in score from clinic visit one (−12 months) to video assessment was 2.0 (2.8SD). Conclusion: The results from the study suggest that video NSAA is partially feasible and reliable.

## 1. Introduction

Duchenne muscular dystrophy (DMD) is a severe, X-linked disease with a worldwide incidence of 1 in 3600 to 6000 live male births [1] and an incidence of 1 in 5136 in Wales following new born screening [2]. Those affected can display mildly delayed motor milestones early in life. The most predominant feature of DMD is progressive proximal muscle weakness and this leads to boys unable to, in most cases, jump, run, or get up from the floor without adopting a Gower’s manoeuvre. Most boys are diagnosed at an average of 4.3 years of age [3] when the functional ability diverges from that of their peers. If not treated with corticosteroids, the agreed stand of care for boys with DMD [4], the disease progresses rapidly with loss of ambulation, on average, at 9 years old [5]. In addition, respiratory, cardiac and orthopaedic complications emerge and without treatment life expectancy is shortened to a mean of 19 years [6].

The standards of care [4] recommend for boys with DMD are assessed once every six months by means of the North Star Ambulatory Assessment (NSAA). The NSAA is a validated and reliable method for assessing the functional ability of boys with DMD [7,8]. It is a 17-item rating scale, with a range of functional tasks such as, stand, run, hop, climb step, get up from the floor, specifically developed by the North Star Clinical Network for Paediatric Neuromuscular Disease to measure function in ambulant children with DMD [9].

There was a global pandemic of the SARS-CoV-2 virus in 2019 which caused severe respiratory illness resulting in imposed restrictions on physical gathering, travelling, leaving house without a reasonable excuse or ‘lockdown’. This ‘lockdown’, in the United Kingdom, was in place from 23 March 2020, and for those with increased risks of contracting the SARS-CoV-2 the Government implemented a 12-week complete isolation protocol. Additionally, a 12-week complete isolation protocol was in place for those classed as ‘extremely vulnerable’ and at increased risk of contracting the virus. Boys with Duchenne muscular dystrophy (DMD) were included in this category and were therefore not able to attend hospital clinic appointments for face-to-face assessments. This lockdown was extended, whereby those patients were discouraged from attending any face-to-face contact unless vital and were advised by the government to shield especially the ones that have been on immunosuppressive treatment like steroids. It was uncertain at the time what future limitations may be in clinical settings with the most vulnerable of the population; however, the necessity will remain to monitor this cohort of patients.

There was a clinical need for NSAA to be assessed to track progression or decline in functional abilities, monitor effectiveness of treatment regimens, and alter glucocorticoid (GC) dosage as required. If functional ability is seen to be deteriorating then in addition to changes in GC dosage several other interventions could be considered and implemented, namely home exercise plans, orthotics, stretching programs or hydrotherapy.

Although the Government has now relaxed SARS-CoV-2 restrictions, the National Health Service has not been able to provide sufficient appoints to reduce the waiting list, now standing at close to 6 million and may be as many as 13 million over the next 4 years [10,11]. Telemedicine, whether this is by phone or video, is a technology that can make it easier for patients to access health care to receive preventative treatments and help with long-term health conditions, such as DMD [12]. This is of particular benefit for those who live at distance from their health care provider, or who face a financial burden of attending [13]. With this background we aimed to conduct this study to investigate feasibility and reliability of NSAA score via a video link.

## 2. Materials and Methods

This study was done as a service evaluation exercise and approval was sought from the clinical governance department of Robert Jones and Agnes Hunt Orthopaedic Hospital, Oswestry, UK (2021_017). Written consent was provided remotely by the parent or guardian of all patients. In addition, parents, or guardians of the DMD boys had previously provided written consent for participation in the NSAA United Kingdom natural history study. Ten boys with a genetically confirmed diagnosis of DMD, who were ambulant and had previous NSAA scores in last 6–12 months were selected from the electronic patient record. Ambulant patients who would be unable to comply with test due to behaviour or learning difficulties were excluded (Appendix A).

The NSAA has seventeen items and a 3-point scoring structure: 2—’normal’—achieves the assessment goal without any assistance or modification; 1—modified method but achieves goal without the need for physical assistance; 0—unable to achieve the goal independently [7,9], meaning a total score can range from: 0, whereby the patient fails all items, to 34, whereby the patient achieves all the activities with no modification of tasks. In addition, the NSAA has the possibility to record two timed items, 10-m timed walk/run test and time to rise from the floor or Gower test [7,9]. The 10-m run/walk and rise from the floor times are not part of the NSAA scoring structure but give an additional measure of physical ability. However, how the boy achieves ‘rise from the floor’ or ‘runs’ are scored and so add to the overall total for the NSAA. A score of zero would be given if the boy were to rise from floor by using an external aid e.g., chair, table or helped by a parent or carer. For the ‘run’ the full 10 m will not be necessary to determine if the boy achieves a ‘run’ or not. A sufficient distance would be required for the boy to accelerate and demonstrate a ‘run’; both feet of ground (no double stance phase) score 2 or pick up speed score or fast walk score 1, or walk (no increase in speed) score 0.

The previous two in-clinic face-to-face NSAA scores were taken as a baseline reference (12 and 6 month) for all patients. This allowed determination of decline or improvement in NSAA scores over a one-year period and served as a standard by which the video scoring could be compared. It has been shown that there is a decline in mean NSAA scores over a 12-month period of −2.2 with a standard deviation of 3.7 [7].

Prior to the test, families were sent information regarding appropriate dress and equipment requirements; including a chair of suitable height (seated with knees and hips at 90 degrees), a step (150 mm in height), and a 10-m walk/run area. The boys were to be barefoot for all items, and in shorts as per NSAA protocol, unless the 10-m run was being performed outside and then shoes/trainers would be worn [9]. The boys selected and consented to the study were familiar with the NSAA, as were their parents, and so required little or no coaching on how to perform the required functional tasks. If part or all the equipment was not available then the percentage of the NSAA completed would be calculated, for example: step not available would give a percentage of 23.5% (8/34) not completed and 8 points be deducted from the NSAA score. This would then be compensated for by increasing the score, for example if the step were missing the score would be 34/(34−8) = 34/26. This method effectively implemented the commonly recommended ‘person mean’ imputation method [14,15].

Two physiotherapists trained and experienced in carrying out and scoring the NSAA performed the assessment. The families were contacted by letter and phone call, and a time and date was arranged for a secure online video link via Attend Anywhere, the National Health Service England platform for conducting video consultations.

During the consultation, the physiotherapists carrying out the video assessment provided guidance of the NSAA protocol for the parents and boys with the assistance of the NSAA protocol information sheet and their tablet computers (iPad, Apple Inc., Cupertino, CA, USA). Once the boy had completed the NSAA, scores were calculated independently by both physiotherapists. This was in keeping with the assessments ordinarily carried out in a face-to-face clinic environment.

The two physiotherapists recorded each item of the NSAA on the score sheet independent of each other and summed the scores to provide a final total out of 34 points. The two physiotherapists were in the same room but did not compare or discuss the scores during or after assessment.

To ensure the correct starting position, families were instructed on positioning of the video camera so that the assessors could ensure this was in accordance with the NSAA protocol.

The remote-assessed score was compared to the two previous scores from clinic appointments. The physiotherapists carrying out the video NSAA were blinded to the previous NSAA scores.

The feasibility of remote scoring was assessed by determining whether items of the NSAA can be performed by parents and their children at home, the technical quality of the videolink and whether using video allowed the physiotherapists to perform the scoring. For this type of study, where the aim is assessing clarity of instructions and ease of administration, a sample size of 10 suffices [16].

Inter-rater reliability of scoring between physiotherapists was assessed by determining the mean absolute difference, a two-way intraclass correlation coefficient (ICC) for agreement and a Bland–Altman plot. The ICC directly measures reliability and can vary from 0 (no reliability) to 1 (perfect reliability). Other measures sometimes used for this purpose, such as the paired t-test or Parson’s correlation coefficient, only assess specific aspects of reliability, whereas the ICC assesses all relevant aspects [17]. The reliability of the clinically assessed NSAA score is very high (ICC = 0.995 [7]). Assuming that the reliability of the remotely assessed score would be at least 0.95, five subjects would suffice to assess the ICC with a 95% confidence interval width of 0.2 [18]. Since a simultaneous clinical assessment of the NSAA was not possible, we evaluated the accuracy of the video-assessed score by gauging if it matched the expected pattern over time of the NSAA based on the two previous clinic-assessed scores. For this we used a linear mixed effects model, using a subject-dependent intercept and either a fixed or subject-dependent slope [19]. Likelihood-ratio tests were used to determine if adding the subject-dependent slope significantly improved the model. Our first measure of matching the expected outcome was the correlation between time of assessment (clinic or video) and NSAA, which we assessed as a repeated measures correlation coefficient based on the mixed model, and using 999 bootstrap samples to determine 95% confidence intervals [20]. Our second measure was to check if the pattern over time in our patients corresponded to what is known, in particular if we saw an increasing rate of function drop with age [21]. We investigated the latter via the interaction of age with slope. For the accuracy analyses, the mean of the two remote assessments was used as the remote assessment score. All statistical analyses were performed using R vs. 4.0.2 (R Foundation for Statistical Computing, Vienna, Austria) using the packages irr lme4 and boot. Two-sided p-values below 0.05 were assumed to denote statistical significance

## 3. Results

The age range of the boys with DMD was 4 years 9 months to 17 years 3 months, with a median of 8 years 2 months. All ten boys were on steroids at the time of in clinic and video NSAA, either prednisolone or deflazacort; nine of the ten boys were on daily dosing and one on intermittent dosing (10 days on 10 days off).

All boys completed the NSAA apart from patient 6 who was unable to perform the rise from floor-timed test as he had on his two previous NSAA in clinic visits (Table 1). The timed 10 m run was only performed by 4 patients, 2 outside with shoes or trainers on and two indoors barefoot (Table 2).

The twenty previous one year and six month NSAA were administered by the same physiotherapist in 75% of clinic visits (15/20). The remaining five assessments were administered by the physiotherapist involved in the video trial. The same two physiotherapists scored the NSAA via video.

The mean time from the first face-to-face clinic visit to video assessment was 10.8 months (range 7 to 19) and from the second face-to-face clinic to video assessment was 5.4 months (range 4 to 7).

All ten video links worked well; the little pixilation on one of the video links did not interfere with scoring the NSAA. No noticeable lag time was present. On three separate occasions, audio was not connected from the attendees’ end but this was easily resolved by either refreshing the link or by exiting the platform and re-entering. The item ‘standing on heels’ proved difficult to assess if the boy was standing on carpet, and therefore any score assessed with the boy on a carpet was lowered by one point. During the assessments, it became clear that the amount of assistance provided by a parent or carer for step climb/descend was difficult to gauge, with the NSAA protocol stating that any assistance is for ‘balance only’. It was therefore decided that any assistance during this part of the NSAA would give the boy a zero score for this item as any score ascribed could not be given with any accuracy.

The two physiotherapists scored all 17 items of the NSAA in all ten boys undertaking the video assessment. Each assessment took between 15 and 20 min in all. The NSAA scores of previous and study visits are shown in Table 1.

The mean absolute difference between the two physiotherapists’ scores was 1.0 point (SD 0.67, range 0–2) and the ICC was 0.98 (95% CI 0.93–1.00). A Bland–Altman plot underlined the small differences between the two physiotherapists and showed no clear pattern (Figure 1).

The mean score for face-to-face clinic visit one was 22.6 points (SD 4.2) and for clinic visit two was 21.8 points (SD 5.3). The mean video-assessed score based on either physiotherapist was 20.6 points (SD 5.7 and 6.5 for physiotherapist 1 and 2, respectively). The mean decline in scores from face-to-face clinic visit one to the video assessment was 1.9 points (SD 3.1) and from the second face-to-face clinic assessment to the video NSAA was 1.1 points (SD 2.4). The mean drop in NSAA over the three assessments was approximately 2.0 points per year (SD 2.8; 95% CI −0.3 to 4.3). Moreover, we found evidence that the rate of drop differed significantly between the children (significant improvement of mixed effect model by including random slope, *p* = 0.019, Figure 2). Based on the model with random slope, the correlation coefficient between assessment time and score was −0.83 (95%CI. 0.66 to 0.89). We also found evidence that the drop increased with age at video-assessment (significant Age*Slope interaction, *p* = 0.018). The drop in score was particularly strong in boys over 8 years old (Figure 1).

Nine boys were able to perform the timed rise from the floor whereas one boy was unable, for all ten boys this was also the case at their two previous clinic visits. The mean absolute difference in rise from floor times between the two video assessments was 0.38 s (SD 0.23, range 0.2–0.9). The ICC for the timed rise was 1.00 (95% CI 1.00 to 1.00).

The timed 10-m run was completed by 4 boys. The video-timed 10-m run showed a mean absolute difference between the two raters of 0.4 s (SD 0.27, range 0.2–0.8), and an ICC of 0.97 (95% CI 0.70–1.00).

## 4. Discussion

With the present pandemic of the SARS-CoV-2 virus and increasing pressure on NHS for face-to-face appointments, it is imperative to find a means for ensuring patients with long-term conditions can be monitored regularly and in keeping with published standards of care. As all worldwide hospitals started to use ‘virtual clinics’ as a means of ‘seeing’ patients, this presented an opportunity to investigate the possibility of using video to assess boys with DMD and investigate the feasibility and reliability of NSAA scores. The results of this study show that video NSAA is partially feasible and reliable, and decline of the score is consistent with published literature.

All families who participated in this study were very co-operative as they provided suitable equipment at home. A variety of set ups were employed; chairs raised up slightly using books, child’s toilet step (exactly 150 mm in height), exercise steps, measuring out a 10-m run, and having the boys dressed appropriately; shorts and barefoot. The study showed that it was possible to score most of the items of NSAA with the video link with good correlation between the two therapists supporting partial feasibility of this assessment. There were however some difficulties, such as a lack of control over types of flooring, some had hard wood floors and some carpets, or difficulties with confirming height of chairs and steps, which may have influenced scoring.

The type of flooring made one aspect of scoring problematic, namely ‘stand on heels’. For families who only had carpeted floors, the ability to ‘stand on heels’ was difficult to assess accurately, so any scores given were scored down regardless of the previous in-clinic scores or likelihood of the boy being able to successfully stand on heels. This obviously may have affected the overall NSAA scores; however, when video scores were compared to previous in-clinic scores any difference was in keeping with the studies conducted by Mazzone et al. [21,22]. In addition, when assistance was provided by a parent for step climb/descend, the assessing physiotherapists could not discern the amount of assistance provided, with the NSAA protocol stating any assistance is for ‘balance only’, the boy would score 0. Despite these limitations, we were able to score almost all seventeen items of NSAA.

The timed tests are an important aspect of the NSAA and perhaps the most contentious aspect of using video due to the potential time lag of Internet connections. However, by starting the stopwatch at the first sign of movement from the boy undertaking a timed functional test and stopping when he achieved the correct finishing position, we tried to remove the effects of any video lag time. This is contrary to the instructions of the NSAA protocol which states the stopwatch should be started on the instruction ‘go’ and any delay in the boy initiating run or rise from floor could be viewed as a decline in their condition. The timed tests appear to be consistent with expected decline, but it would be recommended that any timed tests are treated with caution and only be used as a guide to possible ability and function. It is obvious that far more work needs to be undertaken to confirm the feasibility of performing timed tests via video.

The rise from floor would only be timed if the boy was able to perform the task unaided. He would then be scored for functional ability to perform rise from floor; either 2, or 1 and a time given. A score of 0 would be given as the boy was unable to complete the task due to him requiring external help. Nine out of the ten boys were able to perform the timed rise from floor. One boy could not perform a timed rise from floor but also had been unable to do so in his previous two face-to-face clinic visits (subject 6; age 17 years 3 months). The mean decline for the nine boys in our study able to perform the rise from floor item was 3.8 s per year (SD 7.4). This is similar to the research by Mazzone et al. [22], who reported a mean decline of 5.05 s (SD 12.45).

An obvious problem was the possible variability in the 10 m ‘run’; the distance of which could not be confirmed and used different surfaces (hardwood flooring, carpet, and concrete paths). Although the 10-m distance could not be confirmed, the 4 boys who completed the ‘run’ all had times consistent with face-to-face appointments and expected declines in time to complete the run [21,22]. The same could be suggested for the timed 10-m run but as only 4 boys were able to complete this aspect of the NSAA this would be more difficult to discuss. In spite of the limitation, the run/walk a score of 2 could still be achieved if it is seen that the boy achieves double foot clearance during run (flight; no double stance phase) but no time applied due to not being over a full 10 m.

In spite of these limitations the scores from video seemed consistent with the previous in clinic assessments. Although there was some variability in the scores seen in the video when compared to their face-to-face in clinic visit, this was most likely related to disease progression. A mean decline in NSAA scores of 2.0 points per year in our study is similar to the 2.2 points found in earlier studies [21]. Consistent with earlier studies, we found significantly larger drops in NSAA scores for older boys [21]. The three boys who showed the largest decline in their NSAA scores were aged 10 years 3 months, 17 years and 3 months and 9 years and 4 months, with mean declines of 4 to 5 points per year up to the time of the video assessment. On the other hand, relatively stable or slightly improved scores were seen in all but one of the boys below the age of 8. This is also consistent with data reported earlier, which showed little or no decline in mean scores for children below seven but marked decline in children above seven [21,22].

In this study we demonstrated that remotely assessing the NSAA is partially feasible, and that the NSAA scores obtained had a high inter-rater reliability of 0.98. The mean annual change of 1.9 points over a mean 12-month period is in accordance with the expected change in scores reported by Mazzone et al. [21,22]. However, as this study only recruited 10 boys with DMD, the results would only suggest that remotely scoring the NSAA is reliable and should therefore be treated as a guide to disease progression.

The main limitations of this study are the small sample size, the lack of control over flooring, the inability to confirm the height of steps or the length of 10 m runs and the quality of Internet video links. However, our small sample size was sufficient for the main aims of the study, assessing feasibility and reliability of remotely assessing the NSAA. A more proper validation of our method of home assessment requires a different type of study, where boys are clinically assessed shortly before or after their remote assessment. However, even if the two types of assessment had a slightly lower ICC of 0.9, such a study would still only require a sample size of 15 to assess the ICC with a 95% CI width of 0.2. We therefore suggest that the high intra-rater reliability, combined with the consistency of the NSAA and timed rise from the floor with previous NSAA scores and when compared to larger cohort studies [21,22], would support the use of video assessments as a guide to disease progression. However, we stress that video assessments should be used as a guide to determine if the boy with DMD has lost or maintained function rather than used as a true determinant of ability. The authors at this stage would not recommend for this study to be used for research trials.

One aspect of the study not considered, but important, is the parents’ appreciation of their sons’ assessments. The video assessment reassured them that their sons were still being reviewed and gave them opportunity to discuss any concerns. The familiarity of both the boys and their parents with the physiotherapists helped with the NSAA assessments and allowed for an open discussion and a relaxed atmosphere, even when one or two of the boys were being a little ‘difficult’ and coaxing was required to get best performance. Work by Greenhalgh [23] and colleagues found that virtual clinics worked well when the patient knew the clinician. This was certainly the case with our cohort of patients and therefore it would be advised that video consultations should only be held with patients well known to the neuromuscular service.

The next step will be a comparison of face-to-face in-clinic NSAA to the video assessments to assess its accuracy, which will add further evidence to the usefulness of the video assessments as a means of keeping track of disease progression, allow more frequent assessments if required and save some families long difficult journeys.

## Figures and Tables

**Figure 1 children-09-00728-f001:**
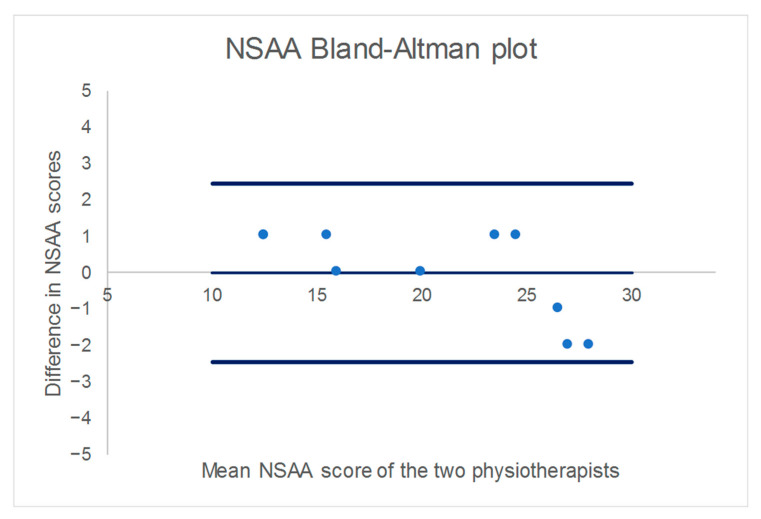
Bland–Altman plot for the NSAA score. The thin blue line represents the bias and the thick blue lines the lower and upper limits of agreement.

**Figure 2 children-09-00728-f002:**
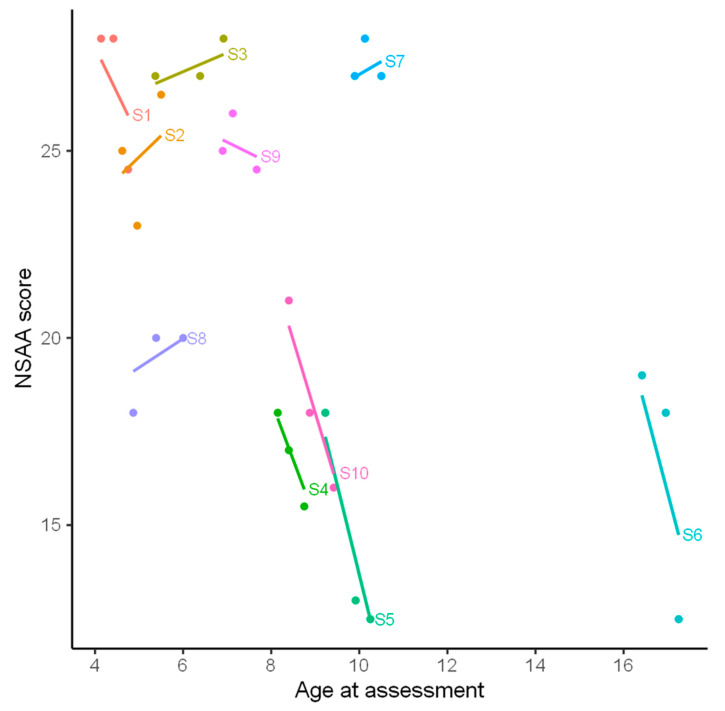
NSAA score versus age at assessment for each of the three assessments of the 10 boys (S1 to S10). The lines are based on a linear mixed model with subject-dependent slope.

**Table 1 children-09-00728-t001:** NSAA and timed rise as collected during the two clinical visits and by the two physiotherapists at a video assessment.

Subject	NSAA Score 1 –12 Months	Timed Rise	NSAA Score 2 –6 Months	Timed Rise	NSAA Video Score P1	Timed Rise P1	NSAA Video Score P2	Timed Rise P2	Mean Score; (P1 & P2)
1	28	5	28	3.5	24	4.2	23	4.7	23.5
2	25	2.7	23	3.2	26	4.2	27	4.0	26.5
3	27	7.7	27	6.0	27	4.2	29	4.4	28
4	18	9.1	17	14.0	16	16.5	15	16.1	15.5
5	18	14.3	13	29.7	13	34.8	12	34.3	12.5
6	19	NA	18	NA	13	NA	12	NA	12.5
7	27	4.3	28	3.9	26	4.00	28	3.6	27
8	18	5.7	20	4.5	20	5.7	20	5.5	20
9	25	4.2	26	3.0	25	3.8	24	2.9	24.5
10	21	6.9	18	10.9	16	15.00	16	15.2	16

P1: physiotherapist 1; P2: physiotherapist 2; NA: data not available.

**Table 2 children-09-00728-t002:** Timed run as collected during the two clinical visits and by the two physiotherapists at a video assessment.

Subject	Clinic Visit 1: Timed Run in Seconds	Clinic Visit 2: Timed Run in Seconds	P1: Time in Seconds	P2: Time in Seconds	Average of P1 and P2 in Seconds
3	5.8 s	5.1 s	5.5 s	5.2 s	5.35 s
5	8.6 s	9.1 s	9.2 s	9.4 s	9.3 s
6	6.4 s	5.6 s	6.1 s	6.4 s	6.25 s
7	5.2 s	5.0 s	5.6 s	4.8 s	5.2 s

P1: Physiotherapist 1 assessment; P2: Physiotherapist 2 assessment.

## Data Availability

SPSS software package.

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
