# Peer review of "Evaluating the Feasibility and Reliability of Remotely Delivering and Scoring the North Star Ambulatory Assessment in Ambulant Patients with Duchenne Muscular Dystrophy"

_children, 2022, doi:10.3390/children9050728_

Round 1

Reviewer 1 Report

The authors sought to evaluate the feasibility and reliability of scoring the NSAA via video link during the Sars-Cov-2 lockdown.  Ten boys with DMD were included in the study with 2 previous in-clinic NSAA scores available.  Remote assessments were scored by 2 PTs independently in an effort to establish the reliability of scoring via this method. 

Unfortunately, the study design as presented does not provide sufficient rigor to support the results and conclusions. While the authors should be commended for continuing to provide care and evaluation during a period of lockdown, the NSAA assessments completed remotely deviated from the standardized protocol in several ways (i.e. inability to view/assess stands on heels due to home environment & parental assist for step items with scores marked down rather than appropriately viewed/scored; mention of partial percentage scores being provided in the instance items were unable to be assessed with no additional percentage scores reported; use of outdoor spaces with shoes donned to complete the 10m in absence of suitable indoor space; deviation of standardized timing for rise from floor and 10m).  Presentation of data that is 'similar' or generally follows the expected decline in ability based on published natural history is not sufficient.  Similarly, the results as reported would not suggest or support the feasibility of performing the NSAA remotely due to the concerns raised above.  The raw inter-rater reliability scoring between the 2 PTs also varied considerably despite high ICCs in this very small cohort; Bland-Altman plots would be useful in evaluating reliability of results to visualize any trends in data variability. 

Author Response

Children-1670399

Status: Pending major revisions

Article type: Article

Title: Evaluating the feasibility of remotely delivering and scoring the North Star Ambulatory Assessment in ambulant patients with Duchenne Muscular Dystrophy

We thank the reviewers for their care in effort in providing feedback on our paper. We have typeset each reviewer’s comment in italic and provided our response in normal font. Reviewer 1 did not number their comments, therefore we separated and numbered them as seemed most appropriate.

Reviewer 1

  1. Unfortunately, the study design as presented does not provide sufficient rigor to support the results and conclusions. While the authors should be commended for continuing to provide care and evaluation during a period of lockdown, the NSAA assessments completed remotely deviated from the standardized protocol in several ways (i.e. inability to view/assess stands on heels due to home environment & parental assist for step items with scores marked down rather than appropriately viewed/scored; mention of partial percentage scores being provided in the instance items were unable to be assessed with no additional percentage scores reported; use of outdoor spaces with shoes donned to complete the 10m in absence of suitable indoor space; deviation of standardized timing for rise from floor and 10m).

We accept that the method by which we assessed the NSAA is not the same as the standardized one. However, it is exactly the essence of our study: demonstrating that one can assess the boys’ ambulatory ability even without being able to follow the standard protocol. Of course, ideally we would have validated our non-standard “home protocol” at a setting that would also allow to perform the standard protocol, but this was not possible at the time. Now that some of our patients can visit us again, we plan to do such a validation study. However, we think it is still worthwhile to present the results we gathered so far.

As for our method of compensating missing items in the score by appropriately scaling the total score, this is simply an application of the “person mean” imputation method. Several studies have demonstrated this is a valid method for scales where at least half the items are present, as was the case in our study (see e.g. Roth et al., 1999; Hawthorne et al., 2005). We have added this explanation to our Methods section.

  1. Presentation of data that is 'similar' or generally follows the expected decline in ability based on published natural history is not sufficient. 

The reviewer is correct that ideally one would want to validate our approach more directly, as e indicate above. However, since that was impossible given the COVID circumstances, we think that our approach, namely comparing to the reasonably expected score, is as good as one could do. In response to the Reviewer’s comments, we have now added extra material to the discussion, which specifically addresses this limitation and suggests a sample size for a validation study comparing directly between the two assessment methods.

  1. Similarly, the results as reported would not suggest or support the feasibility of performing the NSAA remotely due to the concerns raised above. 

We fully understand your concerns, but do feel that perhaps the reviewer risks making perfect the enemy of the good. It would indeed be better to properly validate our home-based remote approach, and we do indeed plan to do so.

  1. The raw inter-rater reliability scoring between the 2 PTs also varied considerably despite high ICCs in this very small cohort; Bland-Altman plots would be useful in evaluating reliability of results to visualize any trends in data variability. 

We have now provided these plots for the NSAA score. Since the differences are small, it is hard to discern a clear trend.

Reviewer 2 Report

This study aimed to investigate the feasibility and reliability of scoring a NSAA for young people with DMD via video link. This study is potentially important in the era of an increased utilisation of telemedicine. Whether or not assessment of NSAA over video link is feasible and reliable is important to understand for the purposes of clinical trial evaluation for potentially life saving therapies for DMD. However, there are significant changes that will be required before this manuscript is suitable for publication. I acknowledge that recruitment for rare diseases, particularly during the COVID-19 pandemic, is difficult. However, the small sample size (n=10) is of concern and the findings from this study need to be interpreted with caution.

Intro:

  1. First paragraph – please provide reference for age at loss of ambulation for steroid-naïve boys.
  2. Paragraph three – suggest more context here that this was a global pandemic, restrictions were placed on movement, social interaction etc. While it can be easily understood at this point in time, years into the future it might be difficult to comprehend what you mean by “SARS-COV-2” and “lockdown”.
  3. Paragraph five – similar to above, suggest more context. Rather than saying “opened up” please provide more context e.g. restrictions on face-to-face contact within healthcare settings have now eased.
  4. Please define NHS.
  5. Paragraph five – please revise sentence one. What do you mean by “appoint” sufficient patients? Do you have any statistics about wait times instead of patient numbers? This might help reader comprehend why this is relevant in the context of standards of care for conducting the NSAA in DMD.
  6. Paragraph five – suggest you mention the burden of travel for families with a son with DMD, considering the physical disability. Suggest include some literature which identifies the competing priorities for families with a son with DMD, time constraints and stress.

Methods:

  1. State which governance department approved the study
  2. Paragraph one – please revise last sentence. You are not excluding non-ambulant patients because of learning difficulties, you are excluding them because it is an assessment for ambulant patients only. Please make it clear that these are two different exclusion criterion.
  3. Please define NHSE
  4. The aims of your study are to investigate the feasibility and reliability of the NSAA via video, however you do not state which outcomes were collected for assessing feasibility. Please also state whether these outcomes were selected a priori.

Results:

  1. There is a lot of detail about the changes in NSAA score over time however this is not stated as an aim of the study. Suggest you focus more on the results for the primary outcomes (feasibility and reliability) and less on trends in NSAA scores. Detailing the outcomes to assess feasibility will assist in structuring the results.

Discussion:

  1. Please consider re-structuring discussion to: main findings of the study, relevance of findings within the context of previous literature and the current health landscape, strengths and limitations, future directions. The first paragraph in particular does not belong in its current location – the first paragraph should state the key findings of the study. Your main objective is to assess the reliability – these findings should be interpreted in the first sections of the discussion.
  2. The small sample size is concerning and places limitations on interpretation. Suggest justifying, with supporting literature, your choice of statistical methods with this sample size.
  3. The interpretation of the key findings for feasibility and reliability assessment are a bit confusing. “Technically feasible” makes it difficult to determine if it is feasible or not. The sentence regarding whether assessing NSAA via video is reliable or not needs revising.

Author Response

Children-1670399

Status: Pending major revisions

Article type: Article

Title: Evaluating the feasibility of remotely delivering and scoring the North Star Ambulatory Assessment in ambulant patients with Duchenne Muscular Dystrophy

We thank the reviewers for their care in effort in providing feedback on our paper.

Reviewer 2

This study aimed to investigate the feasibility and reliability of scoring a NSAA for young people with DMD via video link. This study is potentially important in the era of an increased utilisation of telemedicine. Whether or not assessment of NSAA over video link is feasible and reliable is important to understand for the purposes of clinical trial evaluation for potentially life saving therapies for DMD. However, there are significant changes that will be required before this manuscript is suitable for publication. I acknowledge that recruitment for rare diseases, particularly during the COVID-19 pandemic, is difficult. However, the small sample size (n=10) is of concern and the findings from this study need to be interpreted with caution.

Intro:

  1. First paragraph – please provide reference for age at loss of ambulation for steroid-naïve boys.

Added reference: Peak functional ability and age at loss of ambulation in Duchenne muscular dystrophy - Zambon - - Developmental Medicine & Child Neurology - Wiley Online Library

  1. Paragraph three – suggest more context here that this was a global pandemic, restrictions were placed on movement, social interaction etc. While it can be easily understood at this point in time, years into the future it might be difficult to comprehend what you mean by “SARS-COV-2” and “lockdown”.

Paragraph amended as follows:

This ‘lockdown’, in the United Kingdom, was in place from 23rd March 2020, and for those with increased risks of contracting the SARS-COV-2 the Government implemented a 12-week complete isolation protocol. This imposed restrictions on gatherings and travel-ling, and leaving home without a “reasonable excuse” was prohibited. Additionally, a 12 week complete isolation protocol was in place for those classed as “extremely vulnerable” and at increased risk of contracting the virus. Boys with Duchenne Muscular Dystrophy (DMD) were included in this category and were therefore not able to attend hospital clinic appointments for face to face assessments. This lockdown was extended, whereby those patients were discouraged from attending any  face-to-face contact unless vital and were advised by the government to shield especially the ones that have been on immunosup-pressive treatment like steroids. It was uncertain at the time what future limitations may be in clinical settings with the most vulnerable of the population, however, the necessity will remain to monitor this cohort of patients.

  1. Paragraph five – similar to above, suggest more context. Rather than saying “opened up” please provide more context e.g. restrictions on face-to-face contact within healthcare settings have now eased.

Modified the sentence

  1. Please define NHS.

Done

  1. Paragraph five – please revise sentence one. What do you mean by “appoint” sufficient patients? Do you have any statistics about wait times instead of patient numbers? This might help reader comprehend why this is relevant in the context of standards of care for conducting the NSAA in DMD.

Sentence modified.

  1. Paragraph five – suggest you mention the burden of travel for families with a son with DMD, considering the physical disability. Suggest include some literature which identifies the competing priorities for families with a son with DMD, time constraints and stress.

Williams, K.; Davidson, I.; Rance,M.; Buesch,K.; Acaster, S. Qualitative study on the impact of caring for an ambulatory individual with nonsense mutation Duchenne muscular dystrophy. J Patient Rep Outcomes. 2021,5, 71.

Methods:

  1. State which governance department approved the study

Clinical governance department.

  1. Paragraph one – please revise last sentence. You are not excluding non-ambulant patients because of learning difficulties, you are excluding them because it is an assessment for ambulant patients only. Please make it clear that these are two different exclusion criterion.

This is a typing error, we meant ambulatory patients with behaviour and learning difficulties were excluded.

  1. Please define NHSE

Done

  1. The aims of your study are to investigate the feasibility and reliability of the NSAA via video, however you do not state which outcomes were collected for assessing feasibility. Please also state whether these outcomes were selected a priori.

The outcomes collected for the feasibility were 1) whether items of the NSAA that parents and their children could perform at home, 2) the technical quality of the videolink, and 3) whether using video allowed the physiotherapists to perform the scoring. The outcomes selected for reliability were the inter-rater reliability and the likely accuracy.

Results:

  1. There is a lot of detail about the changes in NSAA score over time however this is not stated as an aim of the study. Suggest you focus more on the results for the primary outcomes (feasibility and reliability) and less on trends in NSAA scores. Detailing the outcomes to assess feasibility will assist in structuring the results.

Thank you for your comment. We have removed most of the detail on the drop in scores, and only left in enough details to demonstrate the accuracy of remotely assessing the NSAA score.

Discussion:

  1. Please consider re-structuring discussion to: main findings of the study, relevance of findings within the context of previous literature and the current health landscape, strengths and limitations, future directions. The first paragraph in particular does not belong in its current location – the first paragraph should state the key findings of the study. Your main objective is to assess the reliability – these findings should be interpreted in the first sections of the discussion.

Thanks, the discussion starts as follows:

With the present pandemic of the Sars-Cov-2 virus and increasing pressure on NHS for face-to-face appointments, it is imperative a means be found to ensure patients with long term conditions can be monitored regularly and in keeping with any published standards of care. As all worldwide hospitals started to use ‘virtual clinics’ as a means of ‘seeing’ patients this presented an opportunity to investigate the possibility of using video to assess boys with DMD and investigate the feasireliability of NSAA scores. The results of this study show that video NSAA is reliable, and decline of the score is consistent with published literature.

  1. The small sample size is concerning and places limitations on interpretation. Suggest justifying, with supporting literature, your choice of statistical methods with this sample size.

The reviewer is correct, the sample size is indeed small. However, it is commensurate with our main aims of assessing feasibility and reliability. We have now added our sample size justification to the Methods section, supported by relevant references, and also in our discussion. Moreover, we have suggested a required sample size for a study aimed at directly validating our remote assessment method by comparing to a clinically-assessed score.

  1. The interpretation of the key findings for feasibility and reliability assessment are a bit confusing. “Technically feasible” makes it difficult to determine if it is feasible or not. The sentence regarding whether assessing NSAA via video is reliable or not needs revising.

Ok we will avoid using terms technical. We agree that this is feasible.

Reviewer 3 Report

I believe this work is potentially important and very useful. This model could clearly lead to an effective way of evaluating and scoring the NSAA in ambulant patients with Duchenne Muscular Dystrophy. However, after reading this article I would suggest some minor edits.

Section 2: Materials and Methods (Last paragraph)

1)Appropriate references for the statistical methods(ICC, Linear mixed effects modeling etc) utilized in this article are missing.

2)A short description on why these particular methods were utilized in comparison to other statistical methods typically utilized for comparing two samples in this field(like t.test, rank sum test etc) may be helpful to the readers who are unfamiliar with this topic. 

Author Response

Children-1670399

Status: Pending major revisions

Article type: Article

Title: Evaluating the feasibility of remotely delivering and scoring the North Star Ambulatory Assessment in ambulant patients with Duchenne Muscular Dystrophy

We thank the reviewers for their care in effort in providing feedback on our paper.

Reviewer 3

I believe this work is potentially important and very useful. This model could clearly lead to an effective way of evaluating and scoring the NSAA in ambulant patients with Duchenne Muscular Dystrophy. However, after reading this article I would suggest some minor edits.

Section 2: Materials and Methods (Last paragraph)

1)Appropriate references for the statistical methods(ICC, Linear mixed effects modeling etc) utilized in this article are missing.

We have now added references for the ICC (which was also used in the original reliability study of the NSAA) and linear mixed models (a review paper describing statistical approaches to study longitudinal studies based on visits – written by a team from the Hospital for Sick Children in Toronto).

2)A short description on why these particular methods were utilized in comparison to other statistical methods typically utilized for comparing two samples in this field(like t.test, rank sum test etc) may be helpful to the readers who are unfamiliar with this topic. 

A statistical test such as t-test or rank-sum test is not appropriate for reliability studies, since one does not want to test a specific research hypothesis (e.g. the mean scores by physiotherapist 1 are different from those by physiotherapist 2) but instead tries to assess the reliability in assessing mobility. The ICC gives a simultaneous assessment of systematic bias and random measurement error. We now describe this advantage.

Round 2

Reviewer 2 Report

Thank for you revising this manuscript. Further changes are required before this manuscript is suitable for publication. 

Abstract/title:

  1. Please be consistent in the wording - in the title you say evaluating the feasibility, in the abstract it's accuracy and reliability and in the body you say feasibility and reliability. Please be clear with the aims of the study, what you measured for each outcome and the results for each outcome. 

Abstract:

  1. Change to "the aim of this study"
  2. If the main outcomes of this study were feasibility and reliability it needs to be stated in the abstract what outcomes were measured for feasibility and the results for this. If the main outcomes were not feasibility then it should not be stated in your title.

Intro:

  1. "SoC" - there is no need to shorten to an acronym if it's only used once in the manuscript
  2. The paragraph about the pandemic still needs more context - please explain clearly in the opening sentence that there was a global pandemic of the SARS-COV-2 virus which caused severe respiratory illness which commenced in 2019 resulting in imposed restrictions on physical gathering or "lockdown". 
  3. "However, the necessity remained to monitor this cohort of patients" sentence is repeated 
  4. "HEP" again, you don't need to define acronym if only used once in the manuscript. Same for HCP.

Methods:

  1. I understand that it would have been a clinical governance department but please state which one (ie. likely it was clinical governance dept from a particular hospital) and also provide the reference number. 

Results:

  1. Table 1 please define “NA”

  1. Figure 2 needs a title

  1. Table 2 – please revise headings. Be consistent with wording of “clinical visit 1…” and “Timed in clinic…”. I’m assuming the physiotherapist times were for the video assessment but please make this clear. Please define what “P1/P2” is.

Discussion:

  1. As above be consistent with your wording of feasibility and reliability

  1. You state “In this study we demonstrated that remotely assessing the NSAA is feasible…” however you were not able to measure all items for: families who had carpet, step climb/descend with parent assistance and 10m run. Please consider revising your statement that video NSAA is feasible considering it is likely that the full NSAA would not be able to conducted for many families.

  1. “However, 342 we stress that but video assessments should be used as a guide to determine if the boy with DMD has lost or maintained function rather than used as a true determinant of ability.” – would you recommend the video NSAA for outcome measures in drug trials? Please include this in your recommendations for practice.

Author Response

Thanks for reviewing the article.

The responses are as follows:

Abstract/title:

Please be consistent in the wording - in the title you say evaluating the feasibility, in the abstract it's accuracy and reliability and in the body you say feasibility and reliability. Please be clear with the aims of the study, what you measured for each outcome and the results for each outcome.

Response: we have corrected this in abstract and title to feasible and reliable.

Abstract:

Change to "the aim of this study"

If the main outcomes of this study were feasibility and reliability it needs to be stated in the abstract what outcomes were measured for feasibility and the results for this. If the main outcomes were not feasibility then it should not be stated in your title.

Response: changed the statements in abstract supporting partial feasibility and reliability.

Intro:

"SoC" - there is no need to shorten to an acronym if it's only used once in the manuscript

Response: deleted

The paragraph about the pandemic still needs more context - please explain clearly in the opening sentence that there was a global pandemic of the SARS-COV-2 virus which caused severe respiratory illness which commenced in 2019 resulting in imposed restrictions on physical gathering or "lockdown".

Response: added this statement:

‘There was a global pandemic of the SARS-COV-2 virus in 2019 which caused severe respiratory illness resulting in imposed restrictions on physical gathering, travelling, leaving house without a reasonale excuse or "lockdown".’

"However, the necessity remained to monitor this cohort of patients" sentence is repeated

Response: Deleted

"HEP" again, you don't need to define acronym if only used once in the manuscript. Same for HCP.

Response: Deleted.

Methods:

I understand that it would have been a clinical governance department but please state which one (ie. likely it was clinical governance dept from a particular hospital) and also provide the reference number.

Response: This study was done as a service evaluation exercise and approval was sought from the clinical governance department of Robert Jones and Agnes Hunt Orthopaedic Hospital, Oswestry, UK (2021_017).

Results:

Table 1 please define “NA”

 Response:

NA: data not available

Figure 2 needs a title

 Response: Tile of both figures is at the bottom of figure. The title for Figure 2 reads as follows:  NSAA score versus age at assessment for each of the three assessments of the 10 boys (S1 to S10).

Table 2 – please revise headings. Be consistent with wording of “clinical visit 1…” and “Timed in clinic…”. I’m assuming the physiotherapist times were for the video assessment but please make this clear. Please define what “P1/P2” is.

Response: amended as follows:

Discussion:

As above be consistent with your wording of feasibility and reliability

You state “In this study we demonstrated that remotely assessing the NSAA is feasible…” however you were not able to measure all items for: families who had carpet, step climb/descend with parent assistance and 10m run. Please consider revising your statement that video NSAA is feasible considering it is likely that the full NSAA would not be able to conducted for many families.

“However, 342 we stress that but video assessments should be used as a guide to determine if the boy with DMD has lost or maintained function rather than used as a true determinant of ability.” – would you recommend the video NSAA for outcome measures in drug trials? Please include this in your recommendations for practice.

Response: The discussion is reviewed and changed with study supporting partial feasibility, reliability. Authors have also added that at this stage the study not ready for research trials.
